# Enhanced YOLOv5: An Efficient Road Object Detection Method

**DOI:** 10.3390/s23208355

**Published:** 2023-10-10

**Authors:** Hao Chen, Zhan Chen, Hang Yu

**Affiliations:** School of Computer and Information Engineering, Tianjin Chengjian University, Tianjin 300384, China; haochen1188@163.com (H.C.); hangyu0508@163.com (H.Y.)

**Keywords:** intelligent traffic, enhanced YOLOv5, multi-scale, road object detection

## Abstract

Accurate identification of road objects is crucial for achieving intelligent traffic systems. However, developing efficient and accurate road object detection methods in complex traffic scenarios has always been a challenging task. The objective of this study was to improve the target detection algorithm for road object detection by enhancing the algorithm’s capability to fuse features of different scales and levels, thereby improving the accurate identification of objects in complex road scenes. We propose an improved method called the Enhanced YOLOv5 algorithm for road object detection. By introducing the Bidirectional Feature Pyramid Network (BiFPN) into the YOLOv5 algorithm, we address the challenges of multi-scale and multi-level feature fusion and enhance the detection capability for objects of different sizes. Additionally, we integrate the Convolutional Block Attention Module (CBAM) into the existing YOLOv5 model to enhance its feature representation capability. Furthermore, we employ a new non-maximum suppression technique called Distance Intersection Over Union (DIOU) to effectively address issues such as misjudgment and duplicate detection when significant overlap occurs between bounding boxes. We use mean Average Precision (mAP) and Precision (P) as evaluation metrics. Finally, experimental results on the BDD100K dataset demonstrate that the improved YOLOv5 algorithm achieves a 1.6% increase in object detection mAP, while the P value increases by 5.3%, effectively improving the accuracy and robustness of road object recognition.

## 1. Introduction

With the emergence of autonomous driving, road detection has become particularly important. However, as the number of cars increases and urban development becomes more complex, there is a greater demand for object detection with higher performance in challenging road conditions. When using cameras or LiDAR to sense the surrounding environment and construct real-time maps, establishing a comprehensive and accurate perception system is key to understanding urban traffic problems. Therefore, precise road object detection has become one of the important issues that need to be addressed.

Most traditional object detection algorithms in the past were based on manually selecting features, which was a labor-intensive task with poor stability. These methods not only fail to deeply interpret image semantics, but also suffer from incomplete feature expressions [1,2]. Deep learning methods have achieved remarkable success in the task of object detection. Among them, two-stage object detection algorithms, such as the Region-based Convolutional Neural Network (R-CNN) series, have demonstrated excellent accuracy. R-CNN first generates candidate region proposals using selective search or other methods, and then extracts features and performs classification on these proposals. Subsequently, Fast R-CNN and Faster R-CNN further improve this process by introducing a Region Proposal Network (RPN) to generate high-quality proposals automatically [3,4]. These methods have achieved high accuracy, especially in scenarios that require precise localization. Another category is one-stage object detection algorithms, such as the You Only Look Once (YOLO) [5] series and the Single Shot MultiBox Detector (SSD) [6,7] algorithm. These methods transform the object detection problem into a regression problem and predict the positions and corresponding classes of objects directly. This end-to-end design enables fast detection speeds, making them suitable for real-time scenarios such as video analysis and edge computing. In addition to deep learning methods, there are also classical machine learning methods applied to object detection tasks. These methods usually use hand-crafted features and traditional machine learning algorithms, such as Support Vector Machines (SVM) [8,9]. Although these methods still have some applications in specific scenarios, they generally have limitations in terms of accuracy and generalization compared to deep learning methods.

Autonomous driving poses great requirements and challenges for the robustness and accuracy of object detection. Object detection in current traffic scenarios usually involves multiple objects with diverse classifications, which increases the difficulty of detection. Moreover, because the position, displacement, shape, and background of the detected targets are uncertain, the images obtained from traffic scenarios often have low resolution due to their complex and variable nature, which increases the difficulty of feature extraction for object detection algorithms. Therefore, the field of autonomous driving is seeking a solution that can achieve high accuracy in different scenarios. Thanks to the application of object detection in the industrial field, especially the deployment on mobile devices, YOLOv5 has been continuously updated [10]. It has obtained lightweight and mature network models, coupled with real-time detection speed, making it very popular on embedded devices such as Jetson. However, YOLOv5 still has problems, such as low confidence in identifying small targets far away, inaccurate positioning, and false alarms, making it difficult to meet the needs of building a complete and accurate perception system for autonomous driving vehicles in complex real-world environments.

To address the problem of multi-scale, multi-level features in object detection tasks that require fusion and enhancement, we introduced the BiFPN improvement method into the YOLOv5 algorithm [11,12]. It uses bidirectional connections to ensure the completeness of feature information and an adaptive weight calculation method for feature fusion, better utilizing features at different scales and levels. We also used DIOU optimization to improve the non-maximum suppression algorithm (NMS) in YOLOv5’s loss function and better evaluate the similarity and distance between detected objects to avoid problems such as misjudgment and duplicate detection. In order to enhance focus on important areas, suppress unimportant background information, and improve detection accuracy, we also introduced the CBAM [13] attention module and conducted experiments on the public dataset Bdd100k to compare the performance differences between YOLOv5 improved by BiFPN technology and traditional YOLOv5.

The remaining parts of the article are organized as follows. First, related works and the existing problems in object detection [14,15] are introduced. Then, the YOLOv5 algorithm for road object detection is presented, and the advantages and disadvantages of the benchmark model are discussed. This provides the background and context for the proposed improvements. The enhancements include the introduction of BiFPN technology, the integration of the CBAM attention module, and the application of the DIOU non-maximum suppression technique. The experimental validation section focuses on comparing the performance of the improved YOLOv5 algorithm with the benchmark model in terms of accuracy and robustness. The results are presented and analyzed to showcase the superiority of the proposed approach. Finally, the discussion and recommendations section examines the strengths, limitations, and future directions of the proposed approach. In it, we discuss the advantages of the approach, acknowledge its limitations, and provide suggestions for future research and applications.

## 2. Related Works

### 2.1. Multi-Scale Feature Fusion

Multi-scale feature fusion [16,17] is a commonly used technique in computer vision aimed at improving model performance by integrating features from different scales. It is particularly important in object detection tasks. Traditional neural networks often use fixed-size filters or pooling operations when processing input images. However, this approach may lead to a loss of low-level details or high-level semantic information. To address this issue, multi-scale feature fusion has been introduced into neural network architectures. There are various methods to achieve multi-scale feature fusion. One common approach is to concatenate or overlay feature maps with different scales. In this way, the network can simultaneously utilize information from multiple scales for decision making. Another method is to use a pyramid structure to generate feature maps at different levels and then fuse them together. This enables the capture of details and semantic information at different scales. Through multi-scale feature fusion, models can better adapt to objects or scenes with varying scales and sizes. This is particularly useful for handling objects or scenes with scale variations. Furthermore, multi-scale feature fusion enhances the robustness of the model and improves its performance in complex scenarios. Multi-scale feature fusion is a beneficial technique that improves the performance of computer vision tasks, especially object detection, by integrating features from different scales. It captures details and semantic information at multiple scales and improves the robustness and adaptability of the model.

Multi-scale feature fusion is an important issue in feature extraction because different scales of features have significant meanings for different tasks. However, the traditional top–down Feature Pyramid Network (FPN) [18,19] is often limited by the one-way information flow and cannot make full use of the different scale features, so a more efficient method is needed to solve this problem. The YOLOv5 algorithm uses the PANet [20,21] network for feature fusion. Compared to FPN, PANet adds a bottom–up path aggregation network to achieve bidirectional information flow. However, the PANet network requires more parameters and computing resources, which means that its speed is relatively slow and not very suitable for real-time target detection tasks. Secondly, although PANet uses a bottom–up path aggregation network to improve the efficiency of information flow, if the low-level feature information is not rich enough or has lost some information, this method may bring some negative effects and lead to a decrease in detection accuracy. Therefore, we modified the original PANet structure in the neck layer of YOLOv5 to a BiFPN network for more efficient multi-scale feature fusion.

### 2.2. Attention Mechanism

Attention mechanism [22,23] is a commonly used technique in deep learning to enhance the focus of neural networks on different parts of input data. It simulates the attention mechanism in the human visual system, allowing the network to selectively attend to information relevant to the current task. In traditional neural networks, every input is treated equally, and all features are simultaneously involved in computation. However, in certain tasks, only a small portion of the input may be crucial for the output. Attention mechanisms were introduced to address this issue. Attention mechanisms have achieved significant success in various deep learning tasks. For example, in image classification tasks, attention mechanisms can assist the network in automatically focusing on regions of an image that are relevant to the classification, thereby improving accuracy. Recently, there have been some interesting methods in the state-of-the-art (SOTA) research in the field of object detection. One of them is Selective Attention for Human Identity (SAHI) [24], which aims to further improve the accuracy of object detection. The SAHI technique adopts a selective attention mechanism that focuses on the identity of individuals in an image, thereby achieving excellent performance in person detection. These latest SOTA research works have propelled the development of the object detection field, enhancing the accuracy and performance of detection algorithms. They hold significant value in terms of image semantic understanding, feature representation, and fast detection. Moreover, they are widely applied in various scenarios that require precise object localization.

To enhance the focus on key areas, suppress unimportant background information, and improve detection accuracy, we introduced the CBAM attention module. CBAM considers both the spatial and channel dimensions, calculating their attention weight coefficients through the spatial attention module and channel attention module, and multiplying them with the input feature maps to adaptively refine the input features. The CBAM attention module is added between the feature extraction backbone network and the inference layer, so that the YOLOv5 algorithm can find regions with high feature map weights and pay more attention to important features during inference.

### 2.3. NMS

NMS [25] is a post-processing technique used in object detection. In object detection tasks, a set of candidate bounding boxes is generated to represent regions that may contain objects [26]. However, due to the nature of network outputs and characteristics of objects in images, these candidate boxes often have overlapping regions. The main purpose of NMS is to eliminate redundant overlapping bounding boxes and select the best representative boxes for each object. It is based on a simple principle: among the candidate boxes for the same object, the one with the highest score is most likely to contain the object, while other candidate boxes with high overlap can be considered redundant.

To address the issues of misjudgment and duplicate detection that may occur when there is overlap between bounding boxes in the YOLOv5 algorithm, we adopted the DIOU algorithm to measure the distance and aspect ratio differences between detection boxes. This algorithm provides a more accurate representation of the positional relationships between objects and helps avoid filtering out boxes that are too close to each other. Specifically, the DIoU_NMS algorithm replaces the traditional IoU measurement with the DIoU measurement to compute the intersection-over-union values between detection boxes, taking into account their distance and aspect ratio differences. During the NMS process, the algorithm filters out detection boxes that are either too close to the selected boxes or have a significant difference in scale, based on a predefined threshold. This improves the accuracy and robustness of the NMS algorithm.

## 3. Benchmark Model and Proposed Methods

The YOLO series of algorithms are the earliest single-stage object detection methods that emerged after RCNN, Fast R-CNN, and Faster R-CNN. YOLO is a new framework proposed to improve the real-time performance of object detection. It can achieve a detection speed of 45 frames per second, and its mAP performance index is also far higher than other real-time detection systems. After that, the YOLO algorithm was continuously improved. YOLOv2 optimized the low accuracy problem in v1, improving the precision and speed of multi-object detection; YOLOv3 chose to add multi-scale training and flexibly process input images, improving the accuracy of small object detection in v2; YOLOv4 solved the problem of GPU training; and YOLOv5 reduced the model size, making it feasible for deployment on mobile edge devices. In this paper, we use YOLOv5 for object detection on an intelligent networked car host, so we will introduce it in detail using YOLOv5 as an example [27,28,29,30].

### 3.1. Benchmark Model

The YOLOv5 network is mainly composed of four parts: input end, backbone main layer, neck feature fusion layer, and head output layer. The overall framework is shown in Figure 1.

The backbone main layer is composed of three parts: a Conv module, a C3 module, and an SPPF module. It is used to extract image features and continually reduce the feature map size. YOLOv5 uses CSPDarknet as the backbone network, which extracts input image features through multiple CBS convolutional layers. After convolution, the C3 module is used for further feature extraction, and the SPPF module performs pooling operations to output feature layers of three scales: 80 × 80, 40 × 40, and 20 × 20.

The Conv module consists of Conv2d, a BatchNorm2d, and a SiLu activation function, mainly used for feature extraction and feature map organization. BatchNorm2d performs batch normalization on batch data, and the SiLu function enhances the non-linear fitting ability of the detection model, as shown in Figure 2.

The C3 module is a feature extraction module that stacks the image features extracted by the CBS convolutional layer to make the feature representation more sufficient. As shown in Figure 3, when the feature map enters C3, it will be processed in two ways. The Conv module in C3 reduces the dimension of the feature map to help the convolution kernel better understand the feature information, and then increases the dimension to extract more complete feature information. Finally, a residual structure is used to extract features, combining the input and output to remove redundant gradient information.

SPPF is an improvement over the SPP spatial pyramid pooling, where feature maps of different scales are converted to the same scale through same pooling. As shown in Figure 4, it combines CBS convolutional layers and three serial 5 × 5 pooling layers to fuse multi-scale features. While further extracting features, it avoids the problem of incomplete expression of deep-level feature information. It extends the region in the input layer corresponding to the point on the feature map, thereby increasing the receptive field of the detection model.

The Neck feature fusion layer obtains shallow image features from the backbone network and concatenates them with deep semantic features to complete the fusion of shallow image features and deep semantic features.

YOLOv5’s neck structure is based on the one-way upsampling FPN structure and expanded to the bidirectional PANet structure, as shown in Figure 5 and Figure 6. By changing the scale of the feature map through interpolation using the upsampling method, the feature map is continuously enlarged to fuse the image features in the backbone network. Different scale feature maps are obtained through downsampling, allowing shallow image features and deep semantic features to complement each other. The neck combines the two paths of different sampling methods to stack the deep features and shallow features of three different scales (20 × 20, 40 × 40, 80 × 80), which are passed layer by layer and finally extracted using the C3 module on the fused features of the three scales, and then passed to the detection layer.

The neck structure effectively fuses shallow and deep features, prevents feature information loss, obtains more complete features, and ensures the feature expression of the detection model for objects of different scales.

The Head output layer is mainly used for detection and consists of the detection module, which includes three 1 × 1 convolutions that correspond to three detection feature layers.

In YOLOv5, the detection layer first divides the three scales of feature maps output by the neck into grids of different scales (80 × 80, 40 × 40, 20 × 20), where each grid corresponds to a pixel, carrying highly condensed feature information. By extracting feature layer information through 1x1 convolutional operations for dimensionality reduction or enhancement, the detection head obtains the position coordinates, categories, and confidence of the anchor in the grid. Then, using anchor boxes of different aspect ratios, the detection layer detects the target object within each grid and adjusts the aspect ratio of the anchor box based on the position information to generate the real box for subsequent detection of position and category information within the box, as shown in Figure 7.

### 3.2. Proposed Methods

Although the original YOLOv5 network performs well in many aspects, there are still some limitations. For example, it may perform relatively poorly in detecting small and dense targets, which can lead to missed or false detections. Additionally, when dealing with targets with a large aspect ratio, such as persons and cars, the detection accuracy is relatively low. Especially during actual driving, the size and resolution of targets in complex and varied real-world scenarios vary greatly, making it a challenge to effectively handle multi-scale features. To address these issues, we propose to improve the YOLOv5 object detection algorithm from three aspects in order to enhance the model’s perceptual ability and detection accuracy. The improved network architecture is shown in the Figure 8.

### 3.3. BiFPN

BiFPN is a novel network structure for multi-scale feature fusion [31,32,33] that addresses the issue of traditional one-way FPN not fully utilizing different scale feature information. BiFPN adds a bottom–up feature path to the FPN and achieves multi-scale feature fusion through bidirectional connections and feature fusion on feature nodes in the feature pyramid network, resulting in improved accuracy and efficiency. Traditional FPN only employs a top–down path for feature fusion in the feature pyramid network, leading to the loss of detailed information in lower-resolution features. BiFPN, on the other hand, captures fine-grained details in low-level features by introducing a bottom–up path and fuses them with high-level features. Additionally, BiFPN enables features to propagate and fuse bidirectionally between different levels through its bidirectional connections, further enhancing feature representation. By effectively utilizing features at different scales and implementing bidirectional connections and feature fusion in the network, BiFPN provides more accurate and efficient feature representations for computer vision tasks such as object detection and image segmentation.

In terms of specific implementation, BiFPN removes nodes with only one input edge and adds extra edges between original input and output nodes to fuse more features. Secondly, BiFPN adds a skip connection. A skip connection is added between the input and output nodes in the same scale, which fuses more features at the same layer without adding too much computational cost. In addition, unlike PANet, which only has one top–down and one bottom–up path, BiFPN considers each bidirectional path as a feature network layer and repeats the same layer multiple times to achieve more advanced feature fusion. The structure of BiFPN is shown in the Figure 9.

#### Weighted Feature Fusion

As different input features have different resolutions, it is crucial to fuse features with different resolutions to improve the accuracy and efficiency of the model. A common method is to adjust features with different resolutions to the same size and then perform addition operation. However, as different features may have different contributions to the output, this method may not achieve the best results. To address this issue, BiFPN proposes a weighted feature fusion method.

BiFPN chooses to add an additional weight for each input feature and lets the network learn the importance of each input feature. To achieve feature fusion, a fast normalization method is adopted, where the weights are divided by the sum of all weights and normalized, as shown in Formula (1):(1)$$O=\sum_{i}\frac{w_{i}}{\varepsilon +\sum_{j}w_{j}}\cdot I_{i}$$

In the formula, *O* represents the output value, *I_i_* represents the input value of a node, *w_i_* represents the weight of the input node, and *j* represents the sum of all input nodes. The condition *w_i_* ≥ 0 is guaranteed by applying the ReLU activation function after each *w_i_*, and ε = 0.0001 is a small value used to prevent numerical instability. Similarly, the values of each normalized weight also fall between 0 and 1. However, since there is no softmax operation involved, the fusion process is more efficient. The final BiFPN integrates both bidirectional cross-scale connections and fast normalized fusion. As an example, we describe here the fusion of two features at level 6 of the BiFPN, as shown in Figure 10.
Figure 10The BiFPN feature fusion process.
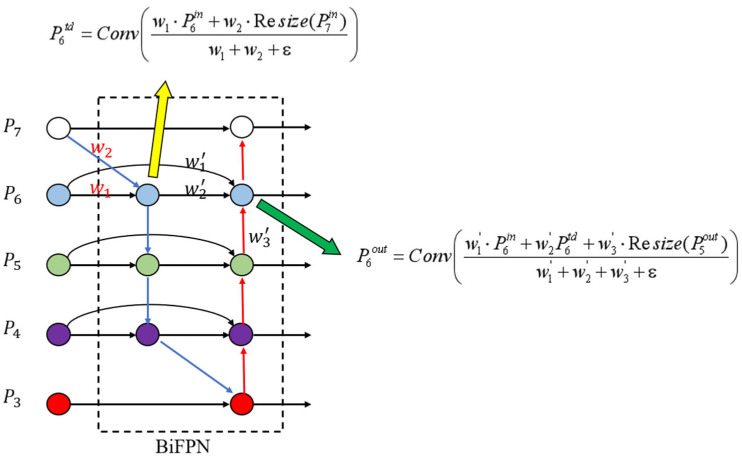

(2)P6td=Convw1⋅P6in+w2⋅Resize(P7in)w1+w2+ε
(3)P6out=Convw1′⋅P6in+w2′P6td+w3′⋅Resize(P5out)w1′+w2′+w3′+ε

Here, *P*_6_*^td^* represents the intermediate feature of the sixth layer in the top–down path, *P*_6_*^out^* represents the output feature of the sixth layer in the bottom–up path, Conv represents the convolution operation, and Resize represents the upsampling or downsampling operation.

### 3.4. CBAM

CBAM [34] is an attention module that enhances the discriminative power of deep neural networks by emphasizing key regions and suppressing background information through spatial and channel attention mechanisms. The spatial attention module assigns weights to pixels in the input feature map based on their spatial positions, allowing the model to focus more on the regions of interest. The channel attention module considers the interdependencies between different channels, balancing the importance of relevant channels and disregarding irrelevant ones. By incorporating CBAM modules between the feature extraction backbone and inference layers, the network can adaptively optimize the input features, thereby improving detection accuracy. CBAM has been widely applied in various computer vision tasks. The CBAM attention process is shown in Figure 11.

As shown in Figure 11, CBAM incorporates attention mechanisms in both the channel and spatial dimensions, which not only saves parameters and computational resources, but also facilitates its integration as a plug-and-play module into existing network architectures. The expression of the CBAM attention process is:(4)$$F\prime =M_{c}(F)⊗F$$
(5)$$F\prime \prime =M_{s}(F\prime )⊗F\prime$$

CBAM consists of the input, channel attention, spatial attention modules, and the output. The input is the feature map *F*, which then goes through the channel attention one-dimensional convolution to produce the channel attention vector *M_c_*. The convolutional result is multiplied elementwise with the original feature map to obtain the channel attention output. Subsequently, the spatial attention two-dimensional convolution, denoted as *M_s_*, is applied, and the resulting output is multiplied elementwise with the original feature map.

The specific process of channel attention is as follows: First, the input feature map *F* is sent to the pooling operation, obtaining the global average pooling vector *V* and the global max pooling vector *V*′. These two pooling vectors represent the mean and maximum values of the features in the channel, respectively, which provides a better understanding of the range and interval of the features in the channel. Then, these two vectors are sent to the shared Multi-Layer Perception (MLP) for addition and sigmoid activation, generating the channel attention vector *Mc*. The channel attention mechanism is shown in Figure 12.

The mathematical expression for the channel attention as follows:(6)Mc(F)=σ(MLP(AvgPool(F))+MLP(MaxPool(F))

This formula obtains a channelwise representation of the input feature map *F* through average pooling, processes it through *MLP* activation functions, and finally maps it to the attention weight *Mc* within the range [0, 1] through the sigmoid function (*σ*).

After obtaining the channel attention vector *Mc* in the channel attention mechanism, it needs to be multiplied with the original feature map *F* to obtain the enhanced feature map *F*′. Then, *F*′ is sent to the spatial attention mechanism, which extracts spatial information through max pooling and average pooling operations, and concatenates these two pooling vectors. The concatenated vector is fed into the shared MPL for convolution and sigmoid activation, generating the spatial attention vector *Ms*, as shown in Figure 13. Finally, *Ms* is multiplied with *F*′ to obtain the final feature map *F*″, which contains both channel and spatial attention information and is used for processing in the next layer of the network.

The mathematical expression for the channel attention as follows:(7)$$Ms(F)=\sigma (f^{7\times 7}([AvgPool(F);MaxPool(F)]))$$

This formula performs a max pooling operation on the input feature map *F*, and processes it through shared *MLP* activation functions, and finally maps it to the attention weight *Ms* within the range [0, 1] through the sigmoid function. $f^{7\times 7}$ performs a convolution operation with a kernel size of 7 × 7.

Specifically, the operation of the CBAM [35] module is as follows: First, the channel attention module performs average pooling and max pooling on the entire input feature map to extract feature information. Then, this feature information is passed through two fully connected layers for processing and a sigmoid function is applied to generate a channel attention weight in the range of 0 to 1. This channel attention weight is multiplied with the original input feature map, resulting in a set of more representative feature maps. Next, the spatial attention module applies a 7 × 7 convolution to these attention-weighted feature maps to further improve their representational power. Finally, the spatial attention-weighted feature maps are multiplied with the original input feature map to obtain feature maps with stronger expressiveness and better classification and detection performance.

### 3.5. DIoU_NMS

DIoU_NMS refers to the method of using DIoU for NMS in object detection. NMS is an important post-processing technique used to filter out redundant detection results and retain the most accurate object detection boxes.

In traditional NMS algorithms, a fixed IoU [36] threshold is set as the suppression threshold, as shown in Figure 14. All detection boxes are sorted by confidence, and all detection boxes are traversed from high to low confidence one by one. The boxes with lower confidence or high IoU overlap with selected boxes are filtered out. However, this method has drawbacks. A very high suppression threshold may filter out some accurate boxes, and only IoU values between boxes are considered, without considering their distance and proportional relationships. In real-world scenarios, overlapping objects may occur during detection. If two targets are close together, this will result in high IoU values. After processing with NMS, only one detection box may be retained, leading to missed detections.

The geometric principle of the DIoU metric is shown in Figure 15, where d represents the distance between the centers of two boxes and c represents the diagonal length of the minimum closed bounding box that covers the two boxes.

The mathematical expression of the DIoU metric is:(8)$$DIoU=IoU-{\left(\frac{d^{2}}{c^{2}}\right)}^{\beta }$$

The parameter β controls the penalty weights of *c*^2^ and *d*^2^ in the DIoU metric. When the value of β is infinitely large, the DIoU metric becomes equivalent to the IoU metric. When the value of β is 0, all boxes that have their centers coincide with the center of the box with the highest score are removed.

## 4. Experimental Results and Analysis

### 4.1. Environment and Parameter Settings

In our experiments, we used a Windows 10 operating system and hardware with a 12th Gen Intel(R) Core(TM) i9-12900K CPU and an NVIDIA GeForce RTX 4090 GPU. The deep learning framework used was Pytorch 2.0.0, and the algorithm was compiled using Python 3.10. For training, we set the epoch to 300 and fixed the initial learning rate at 0.01. The input image size was set to 640 by 640, and the batch size was set to 32. For testing, we used a confidence threshold of 0.25 and an IoU threshold of 0.45.

### 4.2. Database

The dataset used in this paper is the BDD100K dataset, which was developed by researchers at the University of California (Berkeley, CA, USA) and released in 2018. BDD100K (Berkeley DeepDrive 100K) [37] is a large-scale autonomous driving visual dataset that contains approximately 100,000 annotated driving scene images and videos. It covers a wide range of real-world driving scenarios, including urban, suburban, and highway scenes, as well as various weather, lighting, and traffic conditions. This makes it highly beneficial for road detection generalization and application in the field of autonomous driving. However, in consideration of practical needs, we selected 10,000 images suitable for urban road scenes from the BDD100K dataset for our experiments. These 10,000 images were divided into training, validation, and test sets in an 8:1:1 ratio and included nine road object categories, as shown in Figure 16 and Table 1.

### 4.3. Performance Evaluation Metrics

For our experiments, we employed the commonly used Precision (*P*), Recall (*R*), and mean Average Precision (*mAP*) as performance evaluation metrics to assess the detection accuracy of our algorithm. The corresponding formulas are shown below:(9)$$Precision=\frac{TP}{TP+FP}$$
(10)$$Recall=\frac{TP}{TP+FN}$$
(11)$$mAP=\frac{1}{n}\sum_{i=1}^{n}AP_{i}$$

Here, *P* refers to the proportion of correctly detected objects among all detected objects, whereas *R* refers to the proportion of correctly detected objects among all ground truth objects. *mAP* is an evaluation metric that comprehensively considers the performance of object detection algorithms across different categories. For each object category, the average precision (*AP*) can be calculated based on confidence scores and IOU thresholds. *AP* represents the area under the precision-recall curve at different IOU thresholds. The *mAP* is then calculated by taking the average *AP* across all object categories, where a higher *mAP* indicates better algorithmic performance across different categories.

### 4.4. Experimental Results

For object detection tasks, Precision-Recall (PR) curves are usually used to evaluate the performance of models. The PR curve can describe the precision performance at different recall rates, which enables us to have a more comprehensive understanding of the model’s quality. Comparing the PR curves before and after improvement can display the actual improvement of model performance brought by the improvement scheme. The horizontal axis in Figure 17 represents the recall rate, and the vertical axis represents the precision rate. We can see that the model performance after improvement has slightly improved at different recall rates, indicating that the improvement scheme is effective in improving model performance.

The addition of BiFPN to PANet aims to effectively address the limitations of PANet in multi-scale object detection and enhance detection accuracy. BiFPN is a bidirectional feature pyramid network that overcomes the insufficient information flow issue in PANet under multi-scale scenarios by introducing bidirectional connections and multi-scale feature fusion. The key characteristics of BiFPN include bidirectional feature propagation and effective multi-scale fusion ability.

First, bidirectional feature propagation allows information to flow freely between different levels, ensuring effective transmission of multi-scale features. This means that fine-grained details from lower levels can propagate to higher levels, while high-level semantic information can also propagate to lower levels, achieving a more comprehensive and rich feature representation. Second, BiFPN achieves effective multi-scale fusion by integrating features from different levels, using upsampling and downsampling operations to construct a unified feature pyramid structure. This multi-scale fusion capability enables the model to capture objects at different scales simultaneously and improve the accuracy of object detection.

By adding BiFPN to PANet, we can fully utilize the information from multi-scale features and enhance the feature representation ability. This helps address the challenge of scale variation in road object detection and improve the recognition of small and occluded objects. The main evaluation metric is mAP@0.5, and the ablation experiments in Table 2 show that modifying the neck layer from PANet to BiFPN in comparison to YOLOv5s led to a 0.8% increase in mAP@0.5. BiFPN, with its bidirectional feature propagation and multi-scale fusion, compensates for the limitations of PANet and significantly improves the accuracy and performance of multi-scale object detection.

Integrating the CBAM attention module into the YOLOv5s+BiFPN framework brings several benefits. The CBAM module is an attention mechanism designed to enhance feature representation ability, playing a crucial role in road object detection. The basic principle of the CBAM module involves utilizing channel attention and spatial attention mechanisms to enhance feature representation ability. The channel attention mechanism learns the correlations between each channel, adaptively selecting and amplifying the channels that are most relevant to object recognition. This allows the network to focus more on channels with rich semantic information, improving feature discriminability. The spatial attention mechanism learns the correlations between different spatial positions, adaptively selecting and emphasizing important spatial locations while suppressing irrelevant background noise. This is particularly important for road object detection, as it enables the network to attend to the details in the target area while disregarding unimportant background information. By leveraging channel and spatial attention mechanisms, the CBAM module improves feature discriminability, thereby enhancing the accuracy of road object detection. By enhancing the correlations between channels, the CBAM module captures essential semantic information in the target area, making it easier for the network to differentiate between different categories of road objects. Additionally, by utilizing spatial attention mechanisms, the CBAM module provides more attention to the target area, reducing background interference and further improving feature discriminability and object detection accuracy. The experimental results also indicate that adding the CBAM attention module improved mAP@0.5 by 0.5% compared to YOLOv5s+BiFPN, demonstrating an enhancement in feature representation ability.

Introducing DIOU Non-Maximum Suppression (NMS) as an improvement to the YOLOv5s+BiFPN+CBAM framework aims to address the limitations of traditional IoU-based NMS and improve the accuracy of road object detection. Traditional IoU-based NMS only considers the degree of overlap between bounding boxes but neglects their geometric distance relationship. This can result in situations where two bounding boxes have equal overlap but significant differences in spatial positions, leading to false detections of multiple objects or repetitive detection of the same object. This is commonly observed in scenarios where objects are close or occluded. DIOU_NMS overcomes these limitations by using DIOU instead of the traditional IoU as the evaluation metric during non-maximum suppression. When selecting bounding boxes for final detection, DIOU_NMS chooses the one with the highest DIOU value as the primary detection and applies a certain threshold to filter out repetitive detections and false positives. The introduction of DIOU_NMS reduces false detection rates and optimizes the final detection results. By considering the geometric distance between bounding boxes, DIOU_NMS accurately assesses the similarity between them, avoiding situations of repetitive detection and false positives. In comparison to YOLOv5s+BiFPN+CBAM, the experimental results show that DIOU_NMS improved mAP@0.5 to 49.4% and precision to 72%. This further enhances mAP@0.5 and precision values, resulting in more accurate and reliable road object detection.

According to the experimental results in Table 3, the algorithm proposed in this paper demonstrates excellent performance in road object detection tasks and outperformed traditional methods such as Faster R-CNN, SSD, YOLOv3, and YOLOv4 under the same experimental conditions [38,39]. These methods have been widely used in previous research and each has its own advantages and limitations. For example, Faster R-CNN adopts two network modules, one for region proposal and the other for object classification and bounding box regression. SSD is a single-stage object detector that directly predicts object categories and bounding boxes by extracting features through multiple convolutional layers. YOLOv3 and YOLOv4 are also single-stage object detectors known for their fast real-time performance. By comprehensively applying improvement measures, we significantly improved the accuracy of our algorithm in road object detection tasks. The experimental results show that our algorithm performed better in evaluation metrics, especially mAP@0.5, verifying the effectiveness of the proposed improvements. It should be noted that although our algorithm surpasses other common road object detection networks, it may still have certain limitations in specific datasets and scenarios. Therefore, when selecting and optimizing algorithms for practical applications, it is necessary to consider the influence of factors such as dataset characteristics, scene complexity, and environmental conditions.

Comparative experiments indicate a significant performance improvement in road object detection tasks with the proposed algorithm, achieved through improved feature fusion, the introduction of attention mechanisms, and optimized non-maximum suppression. This has important implications for real-world applications in areas such as autonomous driving [40,41,42], establishing a foundation for further research and development. We have also compared the YOLOR model with the enhanced YOLOv5 model on the BDD100k dataset. YOLOR may have a slight advantage in terms of accuracy over the enhanced YOLOv5, but it may be slightly slower in terms of speed. EfficientDet may have higher detection accuracy on the dataset, and due to the adoption of the EfficientNet structure, smaller model sizes can achieve larger object detection precision. However, its implementation complexity and computational intensity are relatively high, which our current hardware cannot effectively meet. SWIN Transformers can improve accuracy when dealing with objects of different scales and demonstrate faster inference time per instance. However, compared to other models, it requires more computational resources and longer training time.

### 4.5. Algorithm Verification

The visual comparison of road object detection results based on the YOLOv5 original algorithm and the proposed algorithm (as shown in Figure 18) reveals that the proposed algorithm exhibits higher confidence in detecting objects positioned at the front of the road, and is able to detect more targets located on side roads compared to the original algorithm. This proves that the proposed algorithm can improve upon issues of poor recognition of multi-scale objects and insufficient feature representation. In mixed traffic flow scenarios under varying lighting conditions, the proposed algorithm demonstrates high recognition accuracy and robustness in detecting dense and overlapping targets, as compared to the original algorithm, thus proving that the proposed algorithm can address problems of false detection and repeated detection, while reducing instances of missed detections.

The proposed algorithm takes into account weather factors for road object detection, and its performance in rainy conditions is worth mentioning. The visual results depicted in Figure 19 clearly demonstrate the advantages of the proposed algorithm in rainy conditions. Compared to the original algorithm, the proposed algorithm exhibits higher confidence and a greater ability to detect side-road targets in rainy road scenarios. Moreover, for multi-scale objects, and especially small targets such as distant traffic lights, the proposed algorithm demonstrates superior performance. This further confirms the capability of the proposed algorithm to improve multi-scale object recognition and address deficiencies in feature representation, validating its ability to meet the demands of complex road scene detection.

## 5. Conclusions and Future Work

In this paper, we propose a method based on the YOLOv5 algorithm to improve the accuracy of object detection in traffic scenes for autonomous driving systems. We introduce BiFPN, the CBAM attention module, and the DIOU non-maximum suppression technique to enhance the algorithm performance. BiFPN is utilized for multi-scale and multi-level feature fusion, and we evaluate its effectiveness in propagating features across different levels and its capability in improving feature fusion quality. By incorporating the CBAM attention module, we improved the network’s focus on target regions and its ability to distinguish features. Additionally, the DIOU non-maximum suppression technique is employed to handle overlapping objects and improve detection accuracy. Compared to the YOLOv5s benchmark model, our proposed model achieved a 1.6% increase in mAP and a significant 5.3% increase in P on the BDD100K validation dataset for traffic scenes.

However, there are still further explorations and discussions to be conducted for our proposed method. More comprehensive comparisons with other state-of-the-art road object detection methods such as EfficientDet and CenterNet can be performed to evaluate performance across different metrics such as mAP and localization accuracy. Furthermore, exploring the deployment of our model on intelligent connected vehicle devices and considering the application of our solution in other domains and tasks would be valuable. For example, applying BiFPN and CBAM attention modules to other object detection tasks such as traffic light color detection or lane marking detection can help validate the generality and effectiveness of our approach across different domains. These explorations will contribute to further improvements and a wider adoption of our method, ultimately enhancing object detection performance in autonomous driving systems.

## Figures and Tables

**Figure 1 sensors-23-08355-f001:**
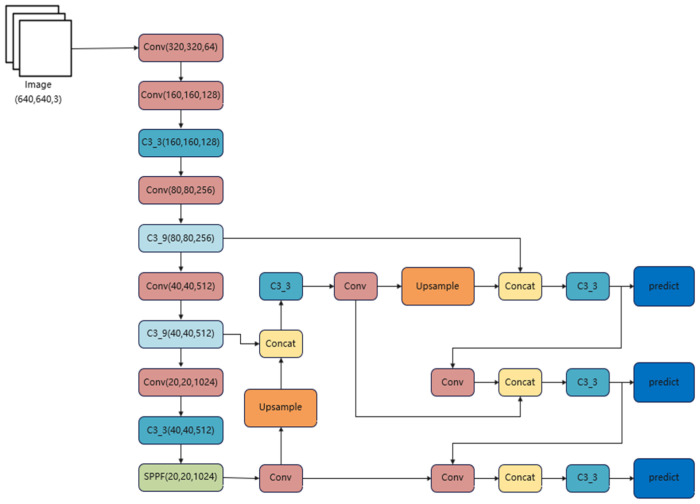
Structural diagram of the YOLOv5 network.

**Figure 2 sensors-23-08355-f002:**
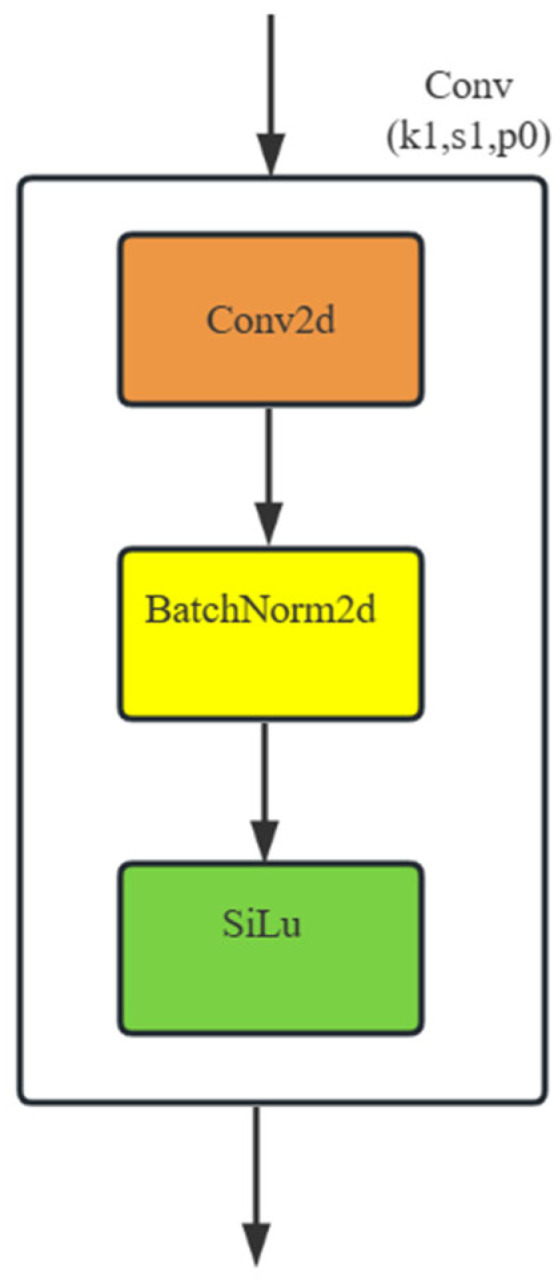
CBS convolutional layer.

**Figure 3 sensors-23-08355-f003:**
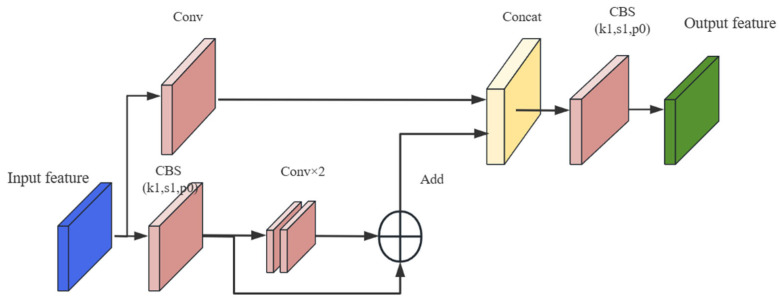
Schematic diagram of the C3 module.

**Figure 4 sensors-23-08355-f004:**
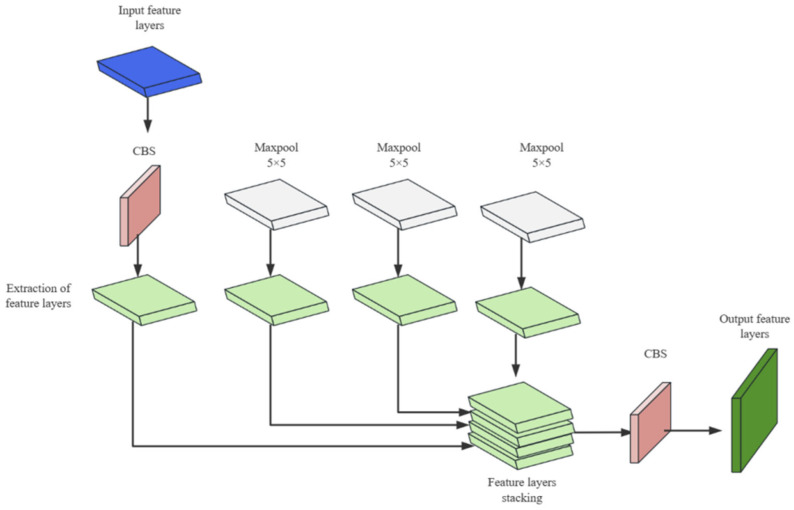
Schematic diagram of the SPPF module.

**Figure 5 sensors-23-08355-f005:**
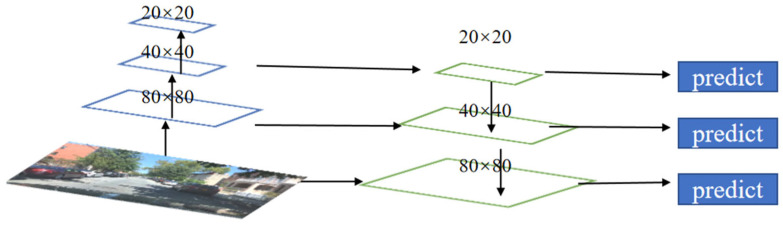
Schematic diagram of the FPN fusion structure.

**Figure 6 sensors-23-08355-f006:**
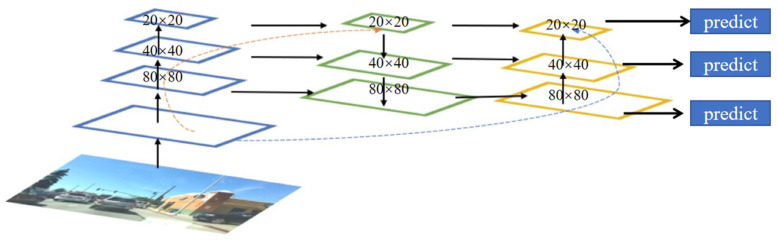
Schematic diagram of the PANet fusion structure.

**Figure 7 sensors-23-08355-f007:**
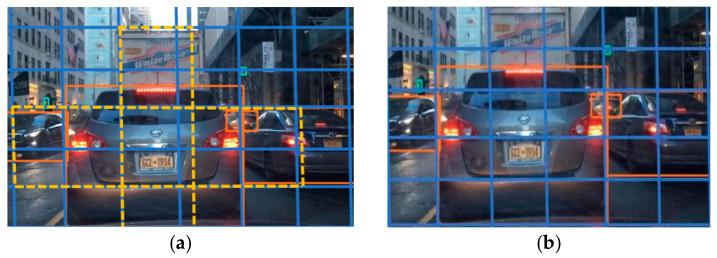
Schematic diagram of the detection process. (**a**) Anchor box. (**b**) Real box.

**Figure 8 sensors-23-08355-f008:**
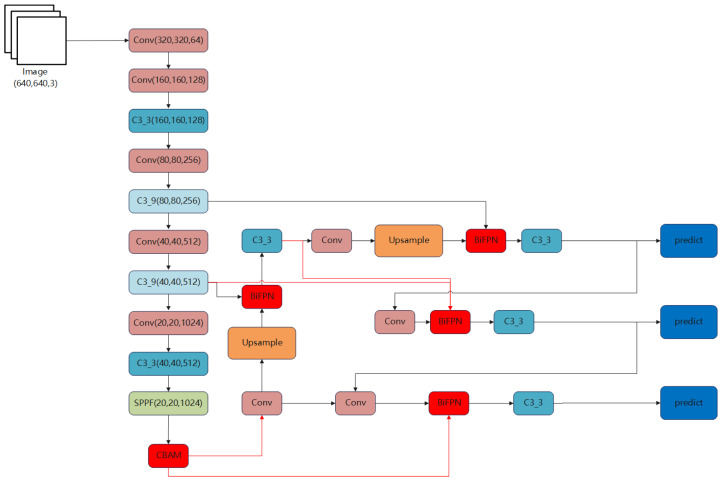
Structural diagram of the improved YOLOv5 network.

**Figure 9 sensors-23-08355-f009:**
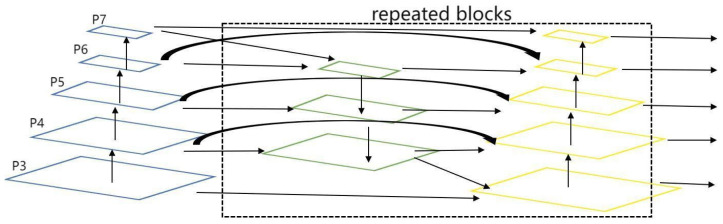
Structural diagram of the BiFPN fusion structure.

**Figure 11 sensors-23-08355-f011:**
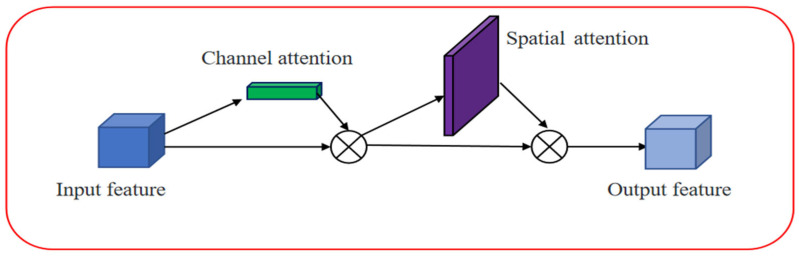
Structural diagram of the CBAM module.

**Figure 12 sensors-23-08355-f012:**
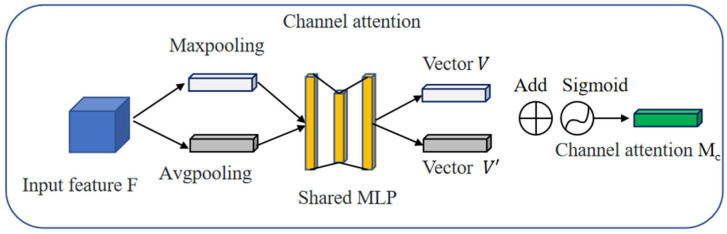
Structural diagram of channel attention mechanism.

**Figure 13 sensors-23-08355-f013:**
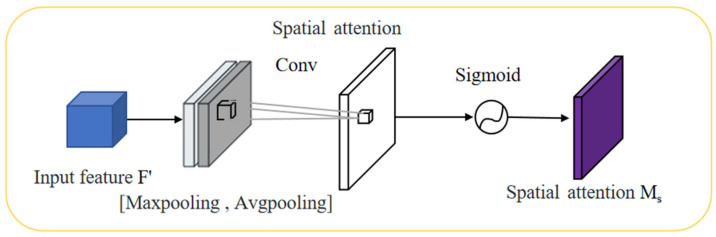
Structural diagram of the spatial attention mechanism.

**Figure 14 sensors-23-08355-f014:**
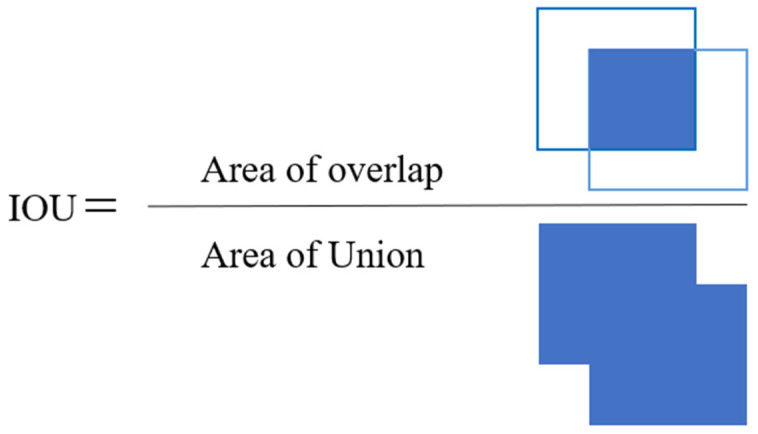
Definition of IOU.

**Figure 15 sensors-23-08355-f015:**
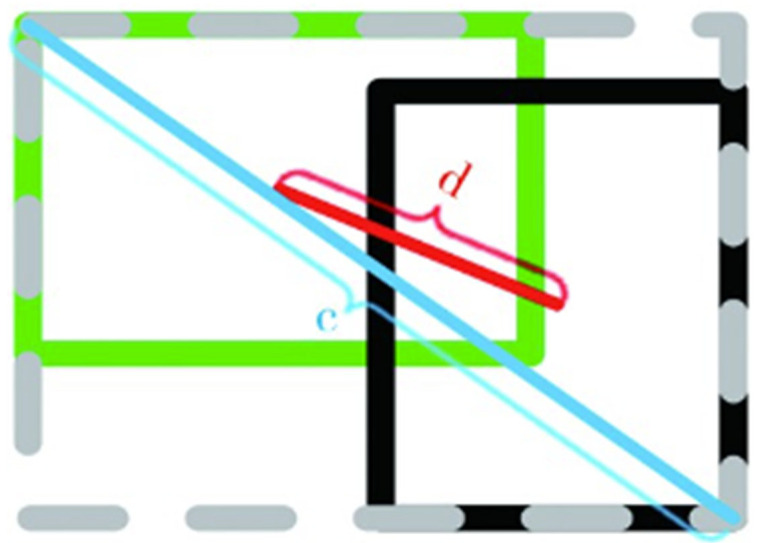
Structural diagram of DIoU.

**Figure 16 sensors-23-08355-f016:**
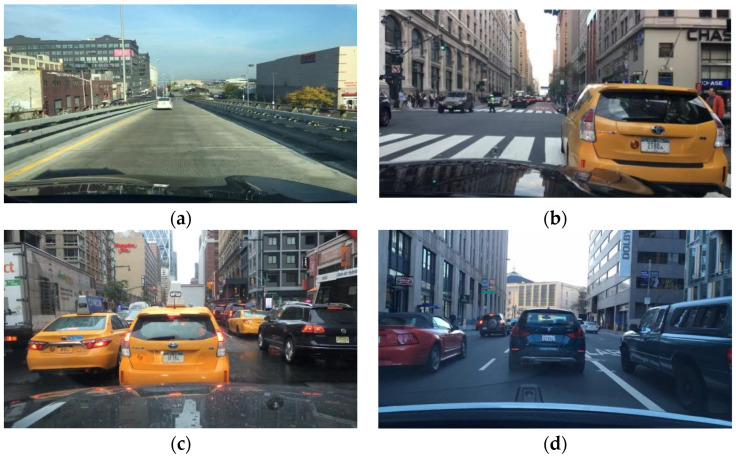
Various road traffic flows in the Bdd100k dataset. (**a**) Free flow. (**b**) Mixed flow. (**c**) Congested flow. (**d**) Slow flow.

**Figure 17 sensors-23-08355-f017:**
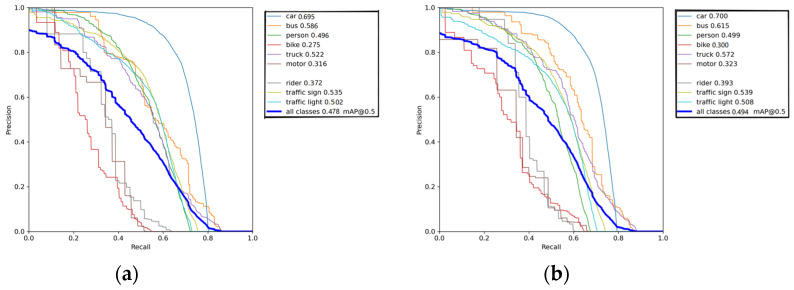
Before and after PR curve comparison. (**a**) Before PR curve comparison. (**b**) After PR curve comparison.

**Figure 18 sensors-23-08355-f018:**
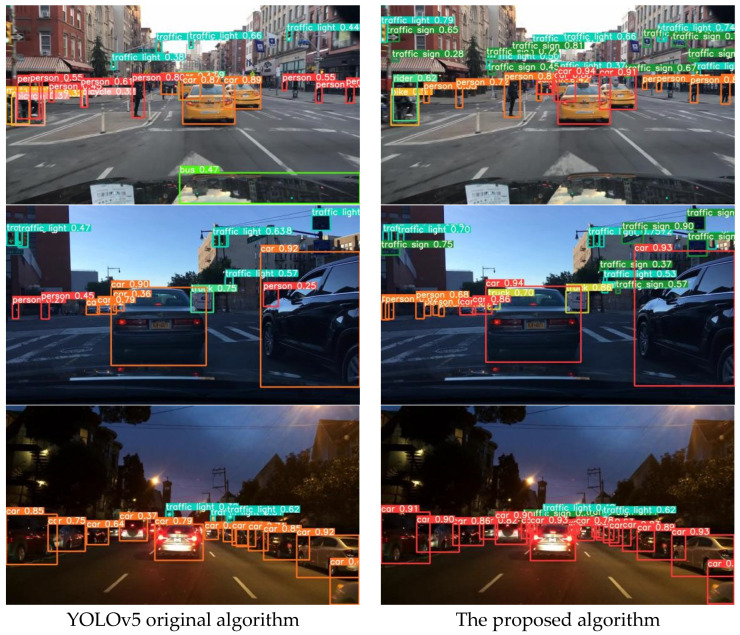
Comparison of road object detection results based on mixed flow in different light conditions.

**Figure 19 sensors-23-08355-f019:**
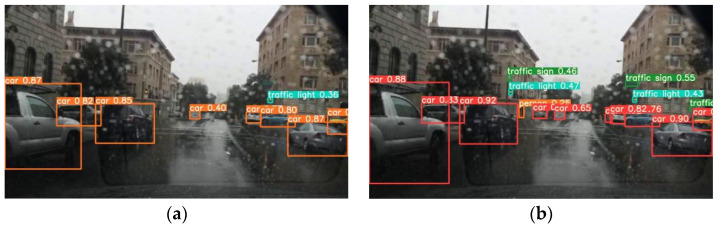
Comparison of road object detection results based on mixed flow in rainy conditions. (**a**) YOLOv5 original algorithm. (**b**) The proposed algorithm.

**Table 1 sensors-23-08355-t001:** The dataset used for the experiment.

Class Name	Train Numbers	Test Numbers
Person	10,614	1199
Rider	512	59
Car	81,881	10,425
Bus	1258	167
Truck	3374	420
Bike	797	92
Motor	364	44
Traffic sign	28,182	3331
Traffic light	21,370	2701

**Table 2 sensors-23-08355-t002:** Ablation experiment results.

Model	Precision (%)	Recall (%)	mAP@0.5 (%)
YOLOv5s	66.7	39.7	47.8
YOLOv5s + BiFPN	69.4	40.5	48.6
YOLOv5s + BiFPN + CBAM	70.5	41.6	49.1
YOLOv5s + BiFPN + CBAM + DIOU_NMS	72	42.1	49.4

**Table 3 sensors-23-08355-t003:** The performance of this algorithm compared with various object detection algorithms.

Algorithm	AP(%)									mAP@0.5(%)	P(%)
	Car	Bus	Person	Bike	Truck	Motor	Rider	Traffic Sign	Traffic Light		
Faster R-CNN	57.4	46.2	30.5	19.7	43.7	19.2	27.4	20.9	8.5	30.4	39.5
SSD	47.3	38.2	18.9	19.6	36.2	18.2	13.7	12	7.4	23.5	67.8
YOLOv3	55.4	44.3	28.9	15.9	42.6	21.6	17.5	29	25.6	31.1	58.9
YOLOv4	60	40.5	50.2	19.2	52	30.4	12.1	48.3	45.5	39.8	41.3
YOLOv5s	69.5	58.6	49.6	27.5	52.2	31.6	37.2	53.5	50.2	47.8	66.7
TheProposed	70	61.5	49.9	30	57.2	32.3	39.3	53.9	50.8	49.4	72

## Data Availability

The dataset used in this study is publicly available, and the corresponding data description is included in the article. Dataset website: https://bdd-data.berkeley.edu/ (accessed on 28 August 2023).

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
