# Peer review of "Enhanced YOLOv5: An Efficient Road Object Detection Method"

_sensors, 2023, doi:10.3390/s23208355_

Round 1

Reviewer 1 Report (Previous Reviewer 2)

I  satisfied with  authors' revision. However, revision is still required for the language including the format and the grammar.

There are numerous grammatical errors in this paper, which do not reach the high standards of the journal of  SENSORS.

Author Response

Dear Reviewer

Thank you for your feedback. We appreciate the reviewers' comments, and we will address the language issues, including formatting and grammar, in our revision. We apologize for any inconvenience caused by these errors and assure you that we will make the necessary improvements to ensure the clarity and accuracy of the manuscript.

Reviewer 2 Report (New Reviewer)

Dear Author, 

This paper is good, but may be it can be significantly improved if You'll check following issues:

1. References list is relevant, and consists of several recent papers (2022,2023) but I've noticed only 16 citations in Your paper. The mos interesting and recent SOTA researches from the ref.list are skipped! Different techniques like SAHI and so on were invented to improve detection accuracy 

2. The performance and accuracy is compared with old versions of YOLO and other networks which are not actual in 2023 (see 1. for actual researches)

Author Response

Dear Reviewer,

1.Thank you for your valuable feedback on our paper. We appreciate your suggestions for improvement and would like to address the issues you have raised.

References List: We apologize for the limited number of citations in our paper. We will carefully review the relevant recent papers from 2022 and 2023, including those you mentioned, to ensure that we include the most interesting and up-to-date state-of-the-art (SOTA) researches in our revised version(Section 2.2). Additionally, we will explore techniques such as SAHI to improve the detection accuracy and incorporate them into our work.

2.Performance and accuracy comparisons: We understand your concerns about comparing our work to outdated versions of YOLO and other networks. However, considering the practical application requirements and stability of our intelligent connected vehicle project, we did not adopt the latest versions widely used in 2023. We will consider incorporating the latest versions of YOLO and other networks in our future research to update the performance and accuracy comparisons, thereby enhancing the relevance and credibility of our study.

Once again, we sincerely appreciate your feedback and assure you that we will make the necessary revisions to address these issues and improve the quality of the paper. Thank you for your time and attention.

Reviewer 3 Report (New Reviewer)

The paper presents a novel road detection detection using extensions in YOLOv5 object detector by including multiscale feature aggregation and attention module.

- The paper needs to be carefully edited to improve readablity. There are instances in the present description that are confusing due to the use of long sentences, e.g., line # 35, "Building a comprehensive and accurate 32 perception system under complex road conditions through cameras or radar laser sensing of the surrounding environment and real-time map construction is the key to solving urban traffic problems." - I believe the author is referring to the understanding of the traffic situation than solving the urban traffic problem itself. 

There are many such examples. A careful reading and editing is required to improve the presentation of the paper. 

- The line #35 referred SVM for road detection without using any reference. Next paragraph, the author switches to the application of deep learning for the problem addressed in this work. There are many other machine learning methods which have heavily studied for road detection. The author should either focus on the use of Deep learning methods in the Introduction or, should highlight other classical machine learning methods applied for the task.

- The literature survey discussed in the paper is very limited. There are methods which has used RCN, SSD and FCN objector detectors for the task. It is advised that the author should mention relevant papers in the literature survey. Some important papers are given below.

- Kim, Huieun, et al. "On-road object detection using deep neural network." 2016 IEEE International Conference on Consumer Electronics-Asia (ICCE-Asia). IEEE, 2016.

- Haris, Malik, and Adam Glowacz. "Road object detection: A comparative study of deep learning-based algorithms." Electronics 10.16 (2021): 1932.

- Roh, Myung-Cheol, and Ju-young Lee. "Refining faster-RCNN for accurate object detection." 2017 fifteenth IAPR international conference on machine vision applications (MVA). IEEE, 2017.

- Section 3 presents the conceptual contribution in the presented work. However, the block diagrams showing different stage of processings is proposed method are not descriptive. The authors should include filter size, padding, normalization types in different blocks in figure 3.1, 3.2, 3.3, 3.5, 3.6 and 3.9. 

- The discussion in Section 3.3 on CBAM can be summarized with appropriate details. The author should also cite relevant paper in the discussion. 

- The reference no 5 and 10 are duplicated. Please resolve the ambiguity.  

- Equations given in Section 3 needs to appropriately discussed.

- The figures are captions are not correctly positioned in the present manuscript. The revised submission needs to make sure that the figures and captions are placed together on the same page.

- The experimental verification of the proposed method establishes the efficacy of the proposed method. The authors should include the citation of relevant papers in the Table 3 to inform the readers about the method used for benchmarking.

- The paper needs to be carefully edited to improve readablity. There are instances in the present description that are confusing due to the use of long sentences, e.g., line # 35, "Building a comprehensive and accurate 32 perception system under complex road conditions through cameras or radar laser sensing of the surrounding environment and real-time map construction is the key to solving urban traffic problems." - I believe the author is referring to the understanding of the traffic situation than solving the urban traffic problem itself. 

There are many such examples. A careful reading and editing is required to improve the presentation of the paper. 

Author Response

Thank you for your review and valuable feedback on the paper. Below are the responses and solutions to each specific issue you raised:

1.Long sentences and readability: We will carefully edit the paper to improve clarity and avoid confusion caused by long sentences. For example, we will break down the long sentence in line 35 into more concise expressions to accurately convey the author's intention regarding understanding traffic situations rather than solving urban traffic problems.

2.Citing relevant machine learning methods: We will focus on introducing the application of deep learning methods in the introduction and mention other classical machine learning methods in the literature review to demonstrate the innovative aspects and differences of the paper.

3.Improving the literature review: We will expand the content of the literature review, including citing the important papers you provided that are relevant to this paper, and cover research that utilizes RCN, SSD, FCN, and other object detection methods.

4.Improving block diagram descriptions: We will add detailed information such as filter size, padding, normalization types, etc., to the blocks in Figures 3.1, 3.2, 3.3, 3.5, 3.6, and 3.9 to increase their informational value and clarity.

5.Enhancing the discussion on CBAM: We will provide more detailed information and cite relevant research papers in the discussion of CBAM.

6.Resolving duplicate references and discussing equations: We will address the issue of duplicate references and provide appropriate discussions and explanations for the equations in Section 3.

7.Placement of figures and captions: In the revised submission, we will ensure that figures and captions are correctly paired and positioned on the same page.

8.Experimental verification and citation: We will include citations of relevant research papers in Table 3 to inform readers about the benchmarking methods used.

9.English language quality: We will thoroughly edit the paper to improve readability, clarity, and eliminate confusion and errors in the English language.

Thank you very much for your review comments. We will revise the paper based on your suggestions to improve its quality and comprehensibility.

Round 2

Reviewer 2 Report (New Reviewer)

Dear Authors,

According to your response: "2.Performance and accuracy comparisons: We understand your concerns about comparing our work to outdated versions of YOLO and other networks. However, considering the practical application requirements and stability of our intelligent connected vehicle project, we did not adopt the latest versions widely used in 2023. We will consider incorporating the latest versions of YOLO and other networks in our future research to update the performance and accuracy comparisons, thereby enhancing the relevance and credibility of our study."

May be my notes were not clearly explicated by me, but I've offered to compare your results with some other projects concerning object detection using contemporary techniques (e.g. simply searching papers with pattern 'YOLO FPN PANet object detection papers with code' for new methods which outperfoms both original YOLO and some descends which uses FPN, PANet and so on). It's not a problem of stability or reliability. I do not insist, but the presentation of the results will benefit if you consider your models and models like YOLOR or EfficientDet or SWIN Transformers or other 2022-2023 projects being compared using AP metrics in table 2 and 3

------

Author Response

Thank you very much for your advice and suggestions. We will consider and compare the latest research on YOLO, FPN, PANet and other methods as you suggested. This comparison will provide a more comprehensive view of the performance differences between our model and YOLOR, EfficientDet, and SWIN Transformers. We will strive to improve the presentation of our results in the paper to ensure that our research is appropriately compared and evaluated. Once again, thank you for your valuable feedback.

Reviewer 3 Report (New Reviewer)

The authors have addressed the comments and issued observed in the original submission. The revised submission should be carefully checked by a native speaker. The present form of the paper is ready to be accepted for publication.

Author Response

Thank you for your review and feedback. We greatly appreciate your efforts in helping us improve the quality of our paper. We have carefully addressed all the comments and issues raised in the original submission, and we are glad to hear that the revised submission is ready to be accepted for publication.

We understand the importance of having a native speaker check the paper for language accuracy, and we will ensure that this is done before final submission. Once again, thank you for your time and valuable feedback.

This manuscript is a resubmission of an earlier submission. The following is a list of the peer review reports and author responses from that submission.

Round 1

Reviewer 2 Report

The reviewer suggests rejecting this paper and has some major concerns as follows.

 1. It is not clear what other new contributions that the authors made. If there is, the authors should highlight them in the introduction.

 2. The writing of the paper is not professional and concise. The writing can be vastly improved.

 3. The authors did not explain the technical details about the proposed method. More importantly, there is a lack of discussions and explanations about the proposed method.

 4. There is insufficient theoretical basis.

 5. Some references are outdated.
